Automated evaluation of multiple sequence alignment methods to handle third generation sequencing errors

Rohmer Coralie 1 2 3 4
Touzet Hélène 1 2 3 4
Limasset Antoine antoine.limasset@univ-lille.fr antoine.limasset@gmail.com 1 2 3 4
1 Université de Lille , Lille , France
2 Ecole Centrale de Lille , Lille , France
3 CNRS , Paris , France
4 UMR 9189, CRIStAL , Lille , France
Gillespie Joseph
Electronic publication date: 2024 Sep 20
Publication date: 2024
Volume: 12
Electronic Location ID: e17731
Received 2023 Oct 6; Accepted 2024 Jun 21
Copyright: ©2024 Rohmer et al.
Copyright year: 2024
Copyright holder: Rohmer et al.
License: This is an open access article distributed under the terms of the Creative Commons Attribution License, which permits unrestricted use, distribution, reproduction and adaptation in any medium and for any purpose provided that it is properly attributed. For attribution, the original author(s), title, publication source (PeerJ) and either DOI or URL of the article must be cited.
License URL: https://creativecommons.org/licenses/by/4.0/

Keywords: Long reads, Multiple sequence alignment, Sequencing errors, Heterozygosity, Pacific bioscience, Oxford nanopore, Benchmark

Funding: The Region Hauts-de-France, along with grants from the French National Research Agency ASTER ANR-16-CE23-0001 AGATE ANR-21-CE45-0012 This study was supported by the Region Hauts-de-France, along with grants from the French National Research Agency: ASTER ANR-16-CE23-0001 and AGATE ANR-21-CE45-0012. The funders had no role in study design, data collection and analysis, decision to publish, or preparation of the manuscript.

==============================
Most third-generation sequencing (TGS) processing tools rely on multiple sequence alignment (MSA) methods to manage sequencing errors. Despite the broad range of MSA approaches available, a limited selection of implementations are commonly used in practice for this type of application, and no comprehensive comparative assessment of existing tools has been undertaken to date. In this context, we have developed an automatic pipeline, named MSA Limit, designed to facilitate the execution and evaluation of diverse MSA methods across a spectrum of conditions representative of TGS reads. MSA Limit offers insights into alignment accuracy, time efficiency, and memory utilization. It serves as a valuable resource for both users and developers, aiding in the assessment of algorithmic performance and assisting users in selecting the most appropriate tool for their specific experimental settings. Through a series of experiments using real and simulated data, we demonstrate the value of such exploration. Our findings reveal that in certain scenarios, popular methods may not consistently exhibit optimal efficiency and that the choice of the most effective method varies depending on factors such as sequencing depth, genome characteristics, and read error patterns. MSA Limit is an open source and freely available tool. All code and data pertaining to it and this manuscript are available at https://gitlab.cristal.univ-lille.fr/crohmer/msa-limit.

Introduction

The introduction and widespread adoption of DNA sequencing have been instrumental for biological research for over 50 years. In the last decade, technologies like Illumina, representative of what is called next-generation sequencing (NGS), have not only made sequencing cost-effective but also increased throughput, broadening access to genomic information. However, the technology continues to evolve, with third-generation sequencing (TGS) technologies addressing key limitations of NGS. One major advantage of TGS is the generation of significantly longer reads—ranging from 104 to 105 nucleotides, even reaching into the megabase range. This performance surpasses that of the NGS, which can only read up to 300 nucleotides (Belser et al., 2018; Miga et al., 2020; Hotaling et al., 2021). These extended reads span a majority of genomic repeats, leading to higher-quality genome assembly. Additionally, TGS employs amplification-free protocols that eliminate the GC bias inherent in NGS, thereby offering a more representative genomic profile (Chen et al., 2013; Lan et al., 2015; Browne et al., 2020). However, TGS is not without challenges. It can introduce a high level of noise and primarily suffer from insertion and deletion errors, ranging from 5 to 15% (Delahaye & Nicolas, 2021), as opposed to the mainly substitution-based errors at lower frequencies (from 1% to 0.1%) in NGS (Stoler & Nekrutenko, 2021).

Current computational tools attempt to manage this noise by leveraging redundancy to sift through erroneous bases and accurately represent genomes (Annis et al., 2020). One common approach involves multiple sequence alignment (MSA), a task known for its computational complexity (Wang & Jiang, 1994; Elias, 2006) and a very rich literature addressing this issue in practice. Various strategies exist to construct MSAs from TGS data, including the selection of a “backbone” read as a reference (Au et al., 2012; Hackl et al., 2014; Goodwin et al., 2015), or the use of more robust but computationally intense methods based on partial order graphs (Lee, Grasso & Sharlow, 2002), which was originally introduced to align sets of homologous genes or proteins.

The first application of partial order graphs to TGS reads can be traced back to PBDAGCON, the error correction module of HGAP (Chin et al., 2013). This trend has then been adopted by numerous tools, some of which directly use the POA (Lee, Grasso & Sharlow, 2002) program, such as Nanocorrect (Loman, Quick & Simpson, 2015), while others, like PBDAGCON (Chin et al., 2013), provide their own implementation of partial order graphs for assembly (Koren et al., 2017; Chin et al., 2016; Xiao et al., 2017) and for correction/polishing (Kundu, Casey & Sung, 2019; Ruan & Li, 2020; Bao et al., 2019; Miyamoto et al., 2014; Ye & Ma, 2016; Morisse et al., 2021). Recently, RACON (Vaser et al., 2017) implemented a faster version of POA based on Single Instruction Multiple Data (SIMD), called SPOA, to enhance correction and polishing. RACON has been extensively used to improve numerous published genomes and is integrated into other tools, such as Unicycler (Wick et al., 2017) and Raven (Vaser & Šikić, 2021). Another SIMD implementation of POA dedicated to long reads is available in abPOA (Gao et al., 2020). Moreover, some of these techniques are even used as part of the read sequencing process. For example, Pacific Bioscience High Fidelity Reads (HiFi) (Wenger et al., 2019) are generated by sequencing a region multiple times and creating a consensus sequence using methods similar to Sparc (Ye & Ma, 2016).

We address this gap through a two-fold contribution: Firstly, we introduce MSA_Limit, an automated toolkit designed to benchmark various MSA tools on TGS datasets against a reference sequence. Built on Snakemake (Köster & Rahmann, 2012) and Conda (Grüning et al., 2018) environments, MSA_Limit offers a user-friendly, easily installable, and flexible framework. A detailed description of the pipeline can be found in section ‘The MSA_Limit Pipeline Overview’. Secondly, we present an extensive set of datasets and benchmark a range of MSA tools. These datasets span bacterial, yeast, and human genomes and serve as a comparative baseline for a selection of widely-used MSA tools from various backgrounds: MUSCLE (Edgar, 2004), T-Coffee (Notredame, Higgins & Heringa, 2000), MAFFT (Katoh et al., 2002), Clustal Omega (Sievers et al., 2011), KALIGN (Lassmann & Sonnhammer, 2005), KALIGN3 (Lassmann, 2020), POA (Lee, Grasso & Sharlow, 2002), SPOA (Vaser et al., 2017) and abPOA (Gao et al., 2020). This benchmarking analysis is discussed in section ‘Benchmarking with MSA_Limit’.

The MSA_Limit Pipeline Overview

Overview of the strategy

The primary goal of MSA_Limit is to provide an automated protocol to evaluate MSA tools on TGS reads. Our investigation focuses on the influence of three critical factors on alignment quality and computational efficiency:

• sequencing error profile, encompassing error rate and error types,

• length of the aligned sequences,

• sequencing depth.

Detailed descriptions of these three factors follow.

Sequencing error profile

It includes various error types such as insertions, deletions, and substitutions present at different rate.

Length of aligned sequences

The aligned sequence length is constrained by read length and also depends on the read processing strategy. For example, tools like CONSENT (Morisse et al., 2021) and ELECTOR (Marchet et al., 2020) employ spliting strategies to focus on smaller subsequences, affecting the length of the actual MSA inputs.

Sequencing depth

We explore sequencing depth values ranging from 10x to 200x, covering a wide array of experimental designs and applications. Generally, guidelines advise against low-depth sequencing below 20x. Indeed, using a Poisson distribution to model sequencing depth, we estimate that with 20x depth, several bases would be missed from a gigabase-sized genome (Hozza, Vinař & Brejová, 2015). As a result, most assemblers designate their comfort zone between 30x and 60x (Phillippy, Koren & Walenz, 2020). We also examine higher sequencing depths of 100x and 200x to determine whether increased information from more sequences leads to improved alignments and to assess the ability of MSA tools to handle such large data.

Pipeline inputs

The MSA_Limit pipeline necessitates a set of TGS reads and a reference sequence for input. The reference sequence acts as the ground truth for MSA quality evaluation. It is not involved in MSA construction.

Pipeline steps

By default, the pipeline conducts various experiments using different region sizes and sequencing depths. Each experiment is distinctly identified by a genomic region, sequencing depth, and the MSA tool in use. The comprehensive process comprises the following seven steps. A bird’s-eye view of the pipeline steps is displayed in Fig. 1 completed with a more detailed depiction in Fig. 2.

Figure 1 Global overview of the MSA_Limit pipeline.

Figure 2 Main steps of the MSA_Limit pipeline.

1. Read alignment: Align the complete set of reads against the reference genome using minimap2 (Li, 2018), with preset options based on the nature of the reads (ONT, PacBio).

2. Starting position selection: Select starting positions for genomic regions. By default, 10 random positions are chosen.

3. Genomic region construction: For each starting position, construct genomic regions of varying lengths. By default, MSA_Limit constructs regions with sizes of 100, 200, 500, 1,000, 2,000, 5,000, and 10,000 bases.

4. Read selection: For each region, select a set of reads that satisfy the desired sequencing depth.

5. MSA construction: Compute the MSA for each available MSA tool using each selection of reads.

6. Consensus sequence creation: Derive a consensus sequence from each MSA. The precise definition of the consensus sequence is provided in section ‘Constructing consensus sequences’.

7. MSA evaluation: Evaluate the MSA by computing a series of metrics from the consensus sequence aligned to the reference sequence. Those metrics are described in section ‘Pipeline outputs and evaluation metrics’.

Constructing consensus sequences

For each MSA, a consensus sequence is built, using the DNA IUPAC code. The method considers each column of the MSA independently and applies a selection procedure to determine which IUPAC character represents the column based on the most frequent characters present in the column. This procedure relies on a threshold parameter, indicating the minimal appearance rate for a nucleotide to be included in the consensus sequence. If the most frequent character is a gap, we retain the gap to represent the column in the consensus sequence. Otherwise, we consider possible nucleotides (A, C, G, T) in descending order of frequency. If the most prevalent nucleotide rate exceeds the threshold, we choose this nucleotide for the consensus. If not, we consider the cumulative rate of the first and second nucleotides. If this rate is above the threshold, we select the corresponding IUPAC character. We continue this process by adding the subsequent nucleotide until the threshold is reached. Note that when selecting the next nucleotide, if there is a tie (i.e., the two following nucleotides have the same occurrence), both nucleotides are added to avoid order bias. We display several examples of consensus sequences using different thresholds in Fig. 3.

Figure 3 Consensus sequence examples with thresholds at 70% and 90%.

The MSA has five sequences. The last row is the consensus sequence.

Pipeline outputs and evaluation metrics

Post-execution of a MSA_Limit run, numerous outputs are generated for detailed analysis, including:

• Identity rate: The ratio of positions where the two sequences have strictly identical characters, divided by the consensus size.

• Ambiguous character rate: The ratio of positions in the consensus sequence where multiple characters are possible (any characters other than A, C, G, T, or gap).

• Match rate: The ratio of positions where the two IUPAC codes share a potential nucleotide character. For instance, Y (which represents T or C) and S (G or C) match because both can represent C, but R (G or A) and Y do not match.

• Error rate: Non-matching characters are considered as errors.

• Consensus size: The length of the consensus sequence.

This is done by pairwise sequence alignement using Exonerate (Slater & Birney, 2005) in the exact global alignment mode. Additionally, summary files providing mean and standard deviations of the metrics across different genomic region starting positions are furnished.

Benchmarking with MSA_Limit

Selection of MSA tools

We benchmarked a diverse set of MSA tools chosen based on their widespread use and complementarity.

Progressive alignment methods:.

These tools initially compute pairwise alignments, which are then progressively merged into a final MSA following a guide tree. The methods differ in pairwise alignment computation, clustering algorithms, and guide tree sequence incorporation. We selected the following tools for this category:

• Clustal Omega: A global sequence aligner using fast hierarchical clustering.

• KALIGN and KALIGN3: Tools that blend local matches into global alignment.

• POA, SPOA, and abPOA: Programs utilizing directed acyclic graphs for intermediate MSAs.

Iterative methods:

Initiating from a rudimentary MSA, these tools iteratively refine it. Selected tools are:

• MUSCLE: Employs k-mer counting, progressive alignment, and tree-dependent refinement.

• MAFFT: Uses the fast Fourier transform (FFT) for quick homologous segment detection.

Consistency check methods:

Tools like T-Coffee precompute both local and global alignments for consistency checks before guide tree construction.

Construction of datasets

Simulated datasets

For precision over reference and error profiles, we used simulated datasets created via PBSIM2 (Ono, Asai & Hamada, 2021), utilizing the E. coli K-12 strain as the reference (GenBank accession GCF_000005845.2). The datasets are named by their error types: DEL (deletions only), INS (insertions only), SUB (substitutions only), and MIX. The MIX datasets contain a proportion of 23% substitutions, 31% insertions, and 46% deletions following an ONT error model. For each of these four error types, DEL, INS, SUB and MIX, we generated eight datasets showcasing different error rates, 1, 2, 5, 10, 15, 20, 25, and 30%, giving a total number of 32 datasets.

Real datasets

We handpicked a selection of ONT real datasets based on three criteria: a reliable reference sequence, sequencing depth exceeding 100x, and diverse estimated sequencing error rates. The reference sequence’s credibility is pivotal, as a perfect sequence is elusive. To minimize discrepancies, we chose genomes derived from the same individual. When such a genome was not available, we required that complementary reads from alternative sequencing technologies, Illumina or HiFi, were available for the same individual and assembled those reads to produce a reference genome.

For each dataset, we estimated the sequencing error rate by aligning the ONT reads on the reference genome using minimap2. The list is available in Table 1.

Table 1 Real datasets employed.

	Reference	Error rate	Depth	
E. coli HiFi	Custom HiFi Assembly	17.28%	200x	
E. coli Illumina	Custom Illumina Assembly	16.36%	650x	
BMB Yeast	Custom Illumina Assembly	10.8%	110x	
Human	T2T-CHM13v2.0	6.6%	120x	

E. coli HiFi.

Derived from the ENA’s SAMN13901561 sample, we accessed both ONT (SRR12801740) and HiFi (SRR11434954) reads. The Hifiasm assembler (Cheng et al., 2021) was utilized for reference genome creation from the HiFi reads, yielding a 17.28% sequencing error rate for the ONT reads against this reference sequence.

E. coli Illumina.

This dataset originates from ENA’s sample SAMN10604456 for the strain CFSAN027350, and provides both ONT (SRR8335315) and Illumina (SRR8333590) reads. A custom SPAdes assembly was generated from the Illumina reads to build the reference genome. The sequencing error rate for ONT reads is estimated to be 16.36%.

BMB Yeast.

This dataset utilized the Illumina sequencing ERR1308675 and ONT sequencing ERR4352154. The reference genome was constructed from Illumina reads with SPAdes, resulting in an estimated 10.8% sequencing error rate for the ONT reads.

Homo sapiens.

We collected ONT data from the T2T consortium (Nurk et al., 2022), and used the T2T-CHM13v2.0 reference genome. ONT reads attained a 6.6% sequencing error rate.

General behaviour of all MSA tools

In our initial experiments, we utilized the nine MSA tools described in section ‘Selection of MSA tools’ and the four real datasets referenced in section ‘Construction of datasets’, namely E .coli Hifi, E. coli Illumina, BMB yeast, and Human. We assessed a broad range of genomic region sizes: 100, 200, 500, 1000, 2000, 5000, and 10,000 bases. Additionally, we varied sequencing depths: 10x, 20x, 30x, 45x, 50x, 60x, 100x, 150x, 200x. For each dataset, we took 10 random regions. A total of 22,680 experiments were conducted. Comprehensive results can be found in the data repository http://gitlab.cristal.univ-lille.fr/crohmer/msa-limit along with the corresponding data http://gitlab.cristal.univ-lille.fr/crohmer/msa-limit-data, while a concise summary is provided in Table 2.

Table 2 Preliminary results for a variety of region lengths and sequencing depths, for all MSA tools and all real datasets.

The first part of the table “Consensus sequence identity rate” is constructed as follows. For each dataset, we selected different region lengths (100, 200, 500, 1000, 2000, 5000, and 10,000 bases) and sequencing depths (10x, 20x, 30x, 45x, 50x, 60x, 100x, 150x, 200x), and for each such combination, we have picked 10 regions. Then we ran every MSA tool on each region, resulting in 10 consensus sequences per tool for each length-depth combination. Those 10 consensus sequences were compared to the reference sequence in order to compute the identity percentage. We deduced from this the average consensus identity percentage associated to a tool and a length-depth combination. It allowed us to determine for each tool which are the lower average identity rate (corresponding to the worst case combination) and the highest average identity rate (corresponding to the best case combination) over all combinations. For each dataset and each MSA tool, we indicate the range min–max, where min (resp. max) refers to the lower average identity rate (resp. higher average identity rate). In the two other sections of the table, the execution time (in seconds) and the memory usage (in MB) are computed for 10 regions of length 200 bases and sequencing depth 10x (min) and 10 regions of length 1000 bases and sequencing depth 100x (max).

	Ecoli-Hifi	Ecoli-Illumina	BMB yeast	Human	
Consensus sequence identity rate	
abPOA	95.2	97.9	95.8	97.4	98.8	99.7	99.6	99.9	
clustal o	83.9	89.7	84.8	91.2	92.4	96.2	95.1	98.6	
KALIGN	93.6	98.7	92.9	97.8	97.8	99.7	99.6	100	
KALIGN3	90.1	98.6	90.7	97.4	96.8	99.7	99.4	99.9	
MAFFT	93.1	98.9	94.2	98.4	98.2	99.8	99.5	100	
MUSCLE	95.5	98.6	95.3	97.8	98.7	99.6	99.7	99.9	
POA	93.7	97.0	96.1	97.1	98.9	99.6	99.7	99.9	
SPOA	95.2	98.6	95.9	98.0	98.9	99.8	99.6	99.9	
T-Coffee	96.2	99.3	95.7	98.3	99.1	99.9	99.7	100.0	
Computation time (in seconds)	
abPOA	0.0	0.7	0.0	0.6	0.0	0.5	0.0	0.6	
clustal o	0.4	89.7	0.4	82	0.4	129.2	0.3	100.9	
KALIGN	0.0	3.4	0.0	3.1	0.0	3.5	0.0	3.3	
KALIGN3	0.1	5.4	0.1	5.3	0.1	6	0.1	5.9	
MAFFT	0.3	8.7	0.2	8.5	0.2	9.2	0.2	8.5	
MUSCLE	0.4	173.3	0.3	171.1	0.3	122.6	0.3	70.9	
POA	0.1	44.7	0.1	24.3	0.1	19.8	0.1	15.4	
SPOA	0.0	2.1	0.0	1.8	0.0	1.7	0.0	1.6	
T-Coffee	1.3	8102.9	12.9	7479.5	13.5	8114.3	13.5	8114.3	
Memory usage (in MB)	
abPOA	3.3	41.3	3.2	15.9	3.1	34.5	3.6	34.6	
clustal o	6.0	78.0	5.9	75.5	5.9	76.0	5.8	74.7	
KALIGN	1.8	3.5	1.8	3.5	1.8	3.4	2.2	3.8	
KALIGN3	4.2	8.8	4.2	7.8	4.1	7.6	4.4	8.2	
MAFFT	21.9	37.1	21.8	36.9	21.9	35.6	22.9	37.5	
MUSCLE	77.0	69.8	7.6	71.1	7.7	71.8	17.6	71.5	
POA	2.4	12.2	2.4	11.3	2.4	11.2	3.8	13.5	
SPOA	8.3	42.3	8.8	38.2	8.0	37.3	8.6	38.0	
T-Coffee	46.5	347.7	46.3	442.2	113.2	424.5	107.9	355.6	

Of the nine MSA tools we tested, seven (abPOA, KALIGN, KALIGN3, MAFFT, MUSCLE, POA, and SPOA) processed all datasets effectively. Clustal Omega and T-Coffee were the exceptions. Although Clustal Omega is known for its accuracy with sequences that are evolutionarily related (Sievers et al., 2011), it underperformed in our long-read alignments. It struggled with managing insertions, deletions, and often introduced spurious gaps, likely because of its “Once a gap, always a gap” paradigm. On the other hand, while T-Coffee produced high-quality MSA results, it was notably resource-intensive, being slower and consuming significant memory. We were not able to run it on larger datasets, such as those with sequencing depth 150x or 200x. Therefore, for large regions or deep sequencing, T-Coffee was not feasible. Given the minor accuracy improvement and significant computational cost, its utility is limited for reads datasets. Moving forward, our analysis omitted Clustal Omega and T-Coffee, focusing on the seven other tools, which are more efficient.

Evaluation of MSA quality

We provide a more detailed analysis into the remaining seven MSA tools’ performance across 100 dataset windows, as opposed to the 10 windows used in previous experiments.

Influence of genomic region size

In Fig. 4, it was observed that genomic region size, spanning from 100 to 10,000 bases, had a negligible impact on consensus identity rate across all sequencing depths, deviating by less than 1%. As a result, the following experiments determined the region size at 500 bases, unless stated otherwise.

Figure 4 Effect of region size on the consensus sequence identity rate for Human, BMB yeast, E. coli Illumina, and E. coli HiFi datasets.

Each dataset is evaluated over 100 distinct regions with 100x depth. The X-axis represents region length (in bases) while the Y-axis indicates the identity percentage between the consensus and reference sequences. Mean identity percentages and standard deviations are depicted.

Influence of the sequencing depth

Figure 5 illustrates the correlation between consensus identity rate and sequencing depth. The human dataset, characterized by its low error rate, showed only minor discrepancies between tools. In contrast, all other datasets, whose sequencing error rate is above 10%, exhibited distinct variability in tool performance.

Figure 5 Effect of sequencing depth on the consensus sequence identity rate for Human, BMB yeast, E. coli Illumina, and E. coli HiFi datasets.

The sequencing depth varies from 10x to 200x for E. coli Illumina and E. coli HiFi datasets, and from 10x to 100x for Human and BMB yeast datasets, whose sequencing depth is smaller (see Table 1). Each dataset is evaluated over 100 distinct regions of size 500. The X-axis represents the sequencing depth, while the Y-axis indicates the identity percentage between the consensus and reference sequences. The figures display the mean consensus identity rate along with the standard deviation.

Although there is a general trend indicating that greater depth improves results, this isn’t always the case. The data suggests that after achieving a depth of approximately 50x, further enhancements in most tools become stagnant. Surprisingly, tools like the POA family and MUSCLE sometimes underperform at increased depths. On the other hand, KALIGN, KALIGN3, and MAFFT consistently show improvement. For example, for Hifi E. coli datasets that exceed 50x depth, most tools stabilize within a 97.5% to 98.5% identity range. However, POA stands out, dropping below 96% at these depths. It is worth noting that while POA and SPOA perform exceptionally well at lower depths, KALIGN and MAFFT achieve their best results at higher depths. Such variations highlight that the choice of the optimal tool largely depends on the specific depth context. The considerable standard deviation, approximately 2% in most instances, emphasizes the significant fluctuation in accuracy depending on region selection. The same observations hold for Illumina E. coli and BMB yeast.

Influence of the sequencing error profile

Understanding the error rate impact is essential since it can vary a lot accross employed technologies and datasets. Figure 5 from the previous paragraph already indicated that lower error rates yield higher accuracy consensus sequences. We delve further into this question using simulated data. In Fig. 6, we demonstrate how consensus identity rate varies with sequencing error rate, specifically following an ONT error distribution. All tools, when tested on simulated data, yielded highly accurate consensus sequences when the error rate was below 10%. Beyond this threshold, most tools’ accuracies plunged, with the exception of POA and SPOA that showed resilience against escalating error rates. Our results validate the selection of POA for processing high error rates, reminiscent of early ONT sequencings.

Figure 6 Effect of error rate on the consensus identity rate on a simulated E. coli dataset.

The dataset consists of 100 distinct regions, each of size 500 bases with a depth of 45x. Reads are generated following the MIX model: 23% substitutions, 31% insertions, and 46% deletions. The graph illustrates the mean consensus identity rate and standard deviation relative to the imposed error rate.

Differing TGS techniques exhibit distinct error patterns. Hence, assessing tools against these errors becomes paramount. In Fig. 7, we delve into the consensus identity rate’s response to varying error types: substitution, insertion, or deletion. The type of error highly influences the performance of the MSA methods. Substitutions are easier to rectify, but the POA family struggles with high substitution error rates. Insertion and deletion errors are more challenging, with deletions being slightly more difficult than insertions.

Figure 7 Effect of error type on the consensus identity rate for a simulated E. coli dataset.

This dataset has 100 regions, each 500 bases in size, and a depth of 45x. The types of errors evaluated are substitutions only (SUB), insertions only (INS), and deletions only (DEL). The graph details the mean consensus identity rate and its standard deviation relative to the error rate.

Evaluation of memory usage and execution time

Influence of the genomic region size

In section ‘Evaluation of MSA quality’, we observed that the influence of sequence size on MSA quality is minimal. However, sequence size significantly affects memory and runtime for most algorithms. Figure 8 illustrates the relationship between memory consumption, running time, and region size. Our results indicate that certain tools, like KALIGN3, KALIGN, and MUSCLE, show linear memory growth with increasing sequence size. In contrast, tools like POA and MAFFT exhibit superlinear growth, while abPOA and SPOA display quadratic growth. Runtime patterns also vary, with some tools appearing almost linear (e.g., SPOA, abPOA, KALIGN, KALIGN3) and others showing superlinear growth (e.g., MUSCLE, POA, MAFFT). In practical scenarios, we observe significant performance disparities among the tested tools in terms of both memory usage and CPU time. These observations confirm the rationale behind previous studies (Morisse et al., 2021; Marchet et al., 2020) that favored partition strategies for constructing MSA from multiple short sequences over long ones.

Figure 8 Effect of the region size on memory usage and CPU time for the E. coli HiFi dataset with 100x depth over 100 distinct regions.

The top figure displays the mean maximal memory usage divided by the region size, the middle figure shows the corresponding CPU time, and the bottom figure represents the mean CPU time according on a log scale. Standard deviation is displayed in black. Notably, the memory curves for KALIGN and KALIGN3 overlap.

Influence of sequencing depth

Figure 9 elucidates the impact of sequencing depth on runtime and memory consumption. Surprisingly, depth has a minimal effect on memory usage with MUSCLE being notable exceptions whose memory scale linearly. Runtime-wise most tools behave super linearly according to the available depth, the fastest growth being MUSCLE that almost scale quadratically.

Figure 9 Effect of sequencing depth on memory usage (top) and CPU time (borrom) for the E. coli HiFi dataset across 100 distinct regions of size 500.

The mean runtime is displayed for each sequencing depth on a log scale.

Influence of sequencing error rate

Figure 10 demonstrates that the error rate marginally impacts time performance except for POA and MUSCLE where a high error rate can double the runtime. Memory wise all tools are almost unaffected by the error rate the POA bases methods that display a linear growth according to the error amount.

Figure 10 Effect of sequencing error rate on memory usage (top) and CPU time (bottom) for MIX simulated E. coli dataset across 100 distinct regions of size 500 with a depth of 45x.

The mean CPU and standard deviation are plotted against the error rate.

Addressing diploid genomes and heterozygosity

In previous sections, polyploid genomes were not addressed. Our datasets were composed of genomes that are functionally haploid, including the human CHM13 genome. This particular human genome, despite being diploid by definition, predominantly arises from the loss of the maternal genetic material and duplication of the paternal genetic material post-fertilization, resulting in a homozygous condition with a 46,XX karyotype. This process effectively renders it haploid for analytical purposes. This approach simplifies the analysis by not accounting for allelic variations, which are minimal in such homozygous contexts.

When analyzing heterozygous organisms, relying solely on a single reference sequence can lead to an overestimation of differences between reads and the reference, especially due to heterozygous local variations. Some contemporary methods can generate “polyploid” reference sequences by distinguishing distinct haplotypes through phasing (Garg, Martin & Marschall, 2016). When multiple haplotypes are available, one approach is to assign each read to a specific haplotype and execute MSA_Limit on each haplotype separately. However, this method is not foolproof. Low polymorphism regions can be challenging to differentiate, especially with highly noisy reads. Due to mapability challenges, many reads may be misassigned, skewing the analysis. Fortunately, a significant portion of haplotype variations, such as SNPs (Single Nucleotide Polymorphisms), can be easily encoded in a reference sequence using the IUPAC code.

In this section, we explore diploid genomes to determine if MSA tools can effectively process reads from distinct alleles. For this purpose, we crafted an artificial diploid genome, ensuring precise knowledge of both haplotypes.

Constructing a heterozygous yeast genome

Inspired by the experiments in the nPhase paper (Abou Saada et al., 2021), we combined datasets from homozygous diploid strains of Saccharomyces cerevisiae to simulate heterozygous yeast genomes. To emulate this, we utilized the BMB strain, introduced another strain (CCN), and combined them to create a “heterozygous yeast”. The reference sequences for the diploid genome were crafted by aligning contigs from both strains using minimap2. The alignments were refined with Exonerate, leading to a consensus sequence where IUPAC symbols indicate heterozygous polymorphisms between the two alleles.

Experimental design

We selected 100 windows of length 500 bases, resulting in a total of 290 SNPs between the two alleles. For each window, we ran the MSA tools, generating one consensus sequence per tool, similar to the approach in section ‘Benchmarking with MSA_Limit’. Given the ploidy degree of two, we set the identity threshold for consensus at 70%, accommodating the sequencing error rate of the datasets. In this context, IUPAC symbols in the consensus sequences are intended to represent polymorphisms between the two alleles. We conducted the experiment at three different sequencing depths: 20x, 50x, and 100x.

Results

Our primary objective was to assess the capability of MSA tools in identifying heterozygous SNPs from the read set. To evaluate this, we first checked if the 290 heterozygous SNPs from the reference genome were present in the consensus sequences generated by each tool. Results are presented in Table 3, detailing recall and precision, and in Fig. 11. While most tools are able to display a high recall that improve with depth (≈75% with 20x, ≈90% with 50x, ≈93% with 100x), all methods display very low precision with only T-Coffee with 100x able to be above 50%. This confirms the known fact that de novo genotyping from TGS is a hard problem (Shafin et al., 2021).

Table 3 Recall and precision for sequencing depth 20x, 50x and 100x for diploid yeast.

For each MSA tool, the recall is computed as the number of IUPAC symbols in the consensus sequence corresponding to heterozygous sites, divided by the number of total heterozygous SNPs (290). The precision is the number of IUPAC symbols in the consensus sequence corresponding to heterozygous sites divided by the total number of IUPAC symbols in the consensus sequence.

Recall	
	abPOA	KALIGN	KALIGN3	MAFFT	MUSCLE	POA	SPOA	T-Coffee	
depth 20	0.72	0.76	0.76	0.80	0.81	0.73	0.69	0.78	
depth 50	0.87	0.92	0.92	0.93	0.94	0.88	0.85	0.93	
depth 100	0.90	0.96	0.95	0.96	0.93	0.92	0.86	0.97	
Precision	
	abPOA	KALIGN	KALIGN3	MAFFT	MUSCLE	POA	SPOA	T-Coffee	
depth 20	0.34	0.22	0.10	0.18	0.28	0.26	0.35	0.43	
depth 50	0.44	0.35	0.19	0.29	0.42	0.30	0.46	0.59	
depth 100	0.43	0.43	0.33	0.40	0.45	0.26	0.44	0.64	

Figure 11 Qualitative performances of the different tools with distinct coverage 20 (left), 50 (middle), 100 (right), and a threshold of 70% for diploid yeast.

SNP/correct IUPAC refers to heterozygous SNPs that are accurately identified in the consensus sequence (true positives), SNP/other to heterozygous SNPs that are not found in the consensus sequence (false negatives), and Non SNP/IUPAC to IUPAC characters present in the consensus sequence that do not correspond to SNPs in the reference genome (false positive).

Discussion

From our experiments, several insights emerge. Foremost, Clustal Omega and T-Coffee appear to be the least suitable among the tested tools. As expected, the performance of other tools, namely MAFFT, MUSCLE, KALIGN, KALIGN3, abPOA, SPOA, and POA, enhances as the sequencing error rate diminishes. These tools exhibit commendable performance with contemporary sequencing data, especially when error rates are around 5%, as observed in the recent Human dataset.

A deeper analysis reveals that POA might be superseded by its variants, abPOA or SPOA. KALIGN3, in comparison to its predecessor KALIGN, seems less compelling. MAFFT emerges as a balanced choice, while MUSCLE’s performance is offset by its computational demands. Interestingly, the influence of sequencing depth is not uniform across tools. Tools from the POA lineage are recommended for datasets with lower depths (10x or 20x), whereas MAFFT, KALIGN, and KALIGN3 excel with datasets having depths greater than 50x.

In section ‘The MSA_Limit Pipeline Overview’, we highlighted the inherent limitations of tools when addressing SNPs in diploid genomes. In such scenarios, no tool provides a comprehensive solution, emphasizing the need for specialized tools.

A pressing question arises: do different tools make errors at unique positions, or are there universally challenging patterns? If each tool errs differently, a combined approach could potentially boost accuracy. To investigate this, we developed a “meta-MSA” using consensus sequences from various tools. We then compared the accuracy of this “metaconsensus” with that of individual tool-specific consensus sequences. As shown in Table 4, the metaconsensus often outperforms individual tools in certain scenarios. However, it doesn’t consistently emerge as the top choice. Notably, it excels in datasets with low coverage and high error rates, and consistently outperforms the least effective tool. This implies that a hybrid approach, drawing on the strengths of multiple tools, might enhance accuracy in specific situations. Yet, it is worth noting that the metaconsensus consistently underperforms compared to Tcoffee. A logical next step would be to experiment with various tool combinations to pinpoint the most effective strategies for specific scenarios.

Table 4 We compare the metaconsensus with the best sequence and the worst sequence selected from all consensus sequences obtained from the different tools (MUSCLE, MAFFT, KALIGN, KALIGN3, POA, SPOA, abPOA), as well as against T-Coffee.

This comparison is conducted for each dataset and at various depths. In the columns for the best sequence, we display how often the metaconsensus shows superior (>), equal (=), or inferior (<) identity rate compared to the best sequence, along with by the average difference observed. Similarly, we report the performance of the metaconsensus in comparison with the worst sequence, and with T-Coffee.

E. coli HiFi (estimated sequencing error rate: 17.28%)	
Depth	   	Best sequence		   	T-Coffee	
		>	=	<	Δaverage	Δaverage		>	=	<	Δaverage	
10		100	0	0	1.76	5.80		0	1	99	−1.32	
20		72	11	17	0.52	3.88		0	0	100	−1.29	
50		50	32	18	0.14	2.57		0	0	100	−1.08	
100		59	13	28	0.16	2.43		0	0	100	−0.93	
E. coli Illumina (estimated sequencing error rate: 16.38%)	
Depth		Best sequence	Worst sequence		T-Coffee	
		>	=	<	Δaverage	Δaverage		>	=	<	Δaverage	
10		95	4	1	1.17	6.07		0	0	100	−2.10	
20		73	14	13	0.44	4.77		0	0	100	−1.87	
50		28	28	44	−0.10	3.34		0	0	100	−2.00	
100		18	17	65	−0.23	2.08		0	0	100	−1.83	
BMB yeast (estimated sequencing error rate: 10.8%)	
Depth		Best sequence	Worst sequence		T-Coffee	
		>	=	<	Δaverage	Δaverage		>	=	<	Δaverage	
10		51	29	20	0.17	2.06		0	16	84	−0.64	
20		8	39	53	−0.13	1.12		0	8	92	−0.68	
50		2	39	59	−0.15	0.57		0	14	86	−0.59	
100		0	31	55	−0.16	0.62		0	13	73	−0.58	
Human (estimated sequencing error rate: 6.6%)	
Depth		Best sequence	Worst sequence		T-Coffee	
		>	=	<	Δaverage	Δaverage		>	=	<	Δaverage	
10		15	72	12	0	0.55		0	63	36	−0.32	
20		0	73	26	−0.07	0.39		0	57	42	−0.38	
50		0	72	27	−0.09	0.26		0	58	41	−0.32	
100		0	72	24	−0.08	0.22		0	61	35	−0.31	
Heterozygous yeast	
Depth		Best sequence	Worst sequence		T-Coffee	
		>	=	<	Δaverage	Δaverage		>	=	<	Δaverage	
10		54	24	22	0.18	1.99		0	1	99	−1.08	
20		4	26	70	−0.18	1.02		0	0	100	−1.27	
50		3	30	67	−0.27	0.57		0	0	100	−1.29	
100		4	20	76	−0.30	0.57		0	0	100	−1.19	

Conclusion

We introduced a robust pipeline to assess the proficiency of MSA tools in generating accurate consensus sequences from TGS data. With its user-friendly design, facilitated by Conda and Snakemake, we envision three straightforward purposes for our tool: benchmarking novel methods, aiding users and developers in refining or selecting a method best suited to their needs, optimizing parameters for their chosen methods to fit their data properties.

In addition to this pipeline, we generated a comprehensive benchmark dataset, which allowed us to unveiled some unexpected results. For instance, the popular SPOA doesn’t always emerge as the best, especially at higher depths. The optimal tool can vary based on the error profile and sequencing depth. Our results also confirmed that existing methods struggle to effectively capture heterozygoty, with a mediocre precision. This could suggest potential enhancements by tweaking scoring systems or amalgamating multiple techniques.

Our study lays the groundwork for developing sophisticated MSA techniques specifically designed for TGS traits. Such advancements could reshape tools used for read correction, assembly refinement, and consensus sequence generation in sequencing devices. Our delve into heterozygosity indicates that MSA can help differentiate between noise and authentic genomic bases. It can also retain variants, laying the groundwork for heterozygosity-conscious read correction or direct TGS data phasing.

Additional Information and Declarations

Competing Interests

Author Contributions

Data Availability

The authors declare there are no competing interests.

Coralie Rohmer conceived and designed the experiments, performed the experiments, analyzed the data, prepared figures and/or tables, authored or reviewed drafts of the article, and approved the final draft.

Hélène Touzet conceived and designed the experiments, analyzed the data, prepared figures and/or tables, authored or reviewed drafts of the article, and approved the final draft.

Antoine Limasset conceived and designed the experiments, analyzed the data, authored or reviewed drafts of the article, and approved the final draft.

The following information was supplied regarding data availability:

The msa limit data is available at Gitlab and Zenodo:

- https://gitlab.cristal.univ-lille.fr/crohmer/msa-limit-data.

- https://gitlab.cristal.univ-lille.fr/crohmer/msa-limit.

- Limasset, A. (2024). MSA-Limit [Data set]. Zenodo. https://doi.org/10.5281/zenodo.13353806.

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
