# Peer review of "Automated evaluation of multiple sequence alignment methods to handle third generation sequencing errors"

_PeerJ, doi:10.7717/peerj.17731_

## Round 0.1 · original submission · Major Revisions

Dear Dr. Rohmer and colleagues:

Thanks for submitting your manuscript to PeerJ. I have now received four independent reviews of your work, and as you will see, the reviewers raised some concerns about the research. Despite this, these reviewers are optimistic about your work and the potential impact it will have on research studying informatics and genomics. Thus, I encourage you to revise your manuscript, accordingly, taking into account all of the concerns raised by both reviewers.

While the concerns of the reviewers are relatively minor, this is a major revision to ensure that the original reviewers have a chance to evaluate your responses to their concerns. There are many suggestions, which I am sure will greatly improve your manuscript once addressed.

Please include all of the relevant information in the tables and figure legends to describe your analyses and interpretations. Also, consider expanding your analyses to cover broader scenarios.

Therefore, I am recommending that you revise your manuscript, accordingly, taking into account all of the issues raised by the reviewers. I do believe that your manuscript will be ready for publication once these issues are addressed.

Good luck with your revision,

-joe

Reviewer 1 ·

Basic reporting

no comment

Experimental design

no comment

Validity of the findings

no comment

Additional comments

The authors have created a tool for evaluating multiple sequence alignment methods in the context of third generation sequencing data. Although such methods have been extensively used to analyse this data, no comprehensive evaluation and comparison of the tools is available. This gap in knowledge is addressed by the manuscript.

The evaluation method is clearly described and the performed experiments are thorough. I only have a couple of minor suggestions:

1. The first paragraph of Introduction could also address HiFi reads which are long and have a very low error rate.

2. In section 3.2.2 it would be good to clearly state that all the data sets use ONT reads as input to the MSA methods.

Reviewer 2 ·

Basic reporting

There are custom references created from identical strains in order to limit reference bias. I assume that because E. coli and Yeast are small genomes that they were successfully assembled completely. However it should be explicitly stated that a single contig was produced if this is the case. I ask this because regions where assemblers fail could be informative to the assessment of the tools. If they’re missing, we’d want to know of that limitation.

Table 2 is hard to follow. Specifically I don’t understand how ‘ lower average identity rate (min)’ and ‘higher identity rate (max)’ are calculated. Is it that you’re describing the range of the observations in the Consensus Sequence Identity section of the table (actual min/max) and then for e.g. Computation time you’ve partitioned a tool’s results by the lower/higher consensus sequence identity and then calculated the two sets average? This is my best guess after looking at this for a while and I’m pretty sure I’m wrong. Please consider reworking this table’s description.
I’ll assume I’m just not understanding table 2 but it is still illustrating something informative. There should be some form of call out to help a reader follow which tool (if any) is doing ‘best’ via e.g. an asterisks or bold text. Perhaps a 5th column of tools’ average across all references and bold text would help? Then a reader could quickly see abPOA is fastest across references, or T-Coffee, MAFFT, and KALIGN are most accurate (if that’s what the table is saying).

Small fixes:

Spelling error in sentence "MSA Evaluation: Evaluate the MSA by computing a serie(s) of metrics from the consensus sequence aligned to the reference sequence. Those metrics are described in subsection 2.5."

Figure 4 could perhaps be friendlier with call outs labeling the columns that are detailed. For example, the ‘2nd column’ detail describes the frequencies of the 4th column. Maybe it should be labeled as ‘4th column’ or a label on the column indicating it is the second would be nice. Also, the 70% and 90% thresholds’ produced 2/3rd identical consensus. Perhaps replacing one of the threshold’s 2/3 with details of the Y/T or H/K consensus would be a more prudent use of space.

Most of us don’t have IUPAC codes memorized. Expand the bullet point:
Match rate: The ratio of positions where the two IUPAC codes share a potential nucleotide character. For instance, Y (which represents T or C) and S (G or C) match because both can represent C, but R (G or A) and Y do not match.

I believe the statement “Our datasets primarily consisted of haploid genomes, with the Human genome being an exception.” is technically correct but slightly misleading. “Most CHM genomes arise from the loss of the maternal complement and duplication of the paternal complement postfertilization and are, therefore, homozygous with a 46,XX karyotype” (https://www.science.org/doi/10.1126/science.abj6987) So while CHM13 is diploid and the statement is correct, its homozygous nature makes it effectively haploid and therefore not really an exception.

Experimental design

The authors have created MSA_Limit to benchmark the utility of multiple sequence alignment approaches to manage sequencing errors. They aim to describe the relative strengths and weaknesses of MSA tools in varying conditions such as sequencing depth, genome characteristics (sequence context?), and sequencer error profiles.

There are 9 main tools in question: MUSCLE, T-Coffee, MAFFT, Clustal Omega, KALIGN, KALIGN3, POA, SPOA, abPOA. Which seems like a pretty comprehensive list to me.

No comment.

Validity of the findings

no comment

·

Basic reporting

1) At the end of the abstract, the grammar of the last sentence could improve. Currently, it says “MSA Limit is open source is freely available at gitlab.cristal.univ-lille.fr/crohmer/msa-limit and all presented results and necessary information to reproduce the experiments are available at gitlab.cristal.univ-lille.fr/crohmer/msa-limit.” This can be strengthened by changing it to “MSA Limit is an open source and freely available tool. All code and data pertaining to MSA Limit, and this manuscript, are available at gitlab.cristal.univ-lille.fr/crohmer/msa-limit.”

2) The first sentence in the introduction says, “Over the past two decades, DNA sequencing has revolutionized biological research”. It has been longer than that. In the mid 1970’s DNA sequencing arrived and facilitated some important discoveries. In 1977, Fredrick Sanger started the Chain Termination Method of sequencing DNA, which ushered in a new era of biological research. See the following review for more: Heather JM, Chain B. The sequence of sequencers: The history of sequencing DNA. Genomics. 2016 Jan;107(1):1-8. doi: 10.1016/j.ygeno.2015.11.003.

3) In the second sentence of the introduction, which says “Technologies like Illumina, representative of second-generation sequencing (SGS)…”. I understand that Illumina is being referenced as Second-Generation Sequencing in the context of Third Generation Sequencing. However, the convention is to call it “Next Generation Sequencing”. In fact, Illumina itself refers to its technology as NGS. https://www.illumina.com/techniques/sequencing.html

4) In the third sentence of the introduction, it is written: “third-generation sequencing technologies (TGS)”. The abbreviation should come between “sequencing” and “technologies”. Therefore, all subsequence plural references to TGS should be singular, as it is a singular class of technology. The same is true for NGS. There are different technologies within the NGS class of strategies. NGS is a singular entity.

5) It might be helpful to define the algorithm abbreviations in-text. For example, PBDAGCON, HGAP, POA, RACON, SIMD, SPOA, and others are used without defining the abbreviations.

6) Under section 2.3 Pipeline Steps, the phrase “A bird eye view of the pipeline steps is displayed…”. The phrase should be corrected to “A bird’s-eye view”.

7) In section 2.3 Pipeline Steps, number 5 says, “For each available MSA tools, compute the MSA from each selection of reads”. However, the grammar of this sentence could be improved by using something like “Compute the MSA for each available MSA tool using each selection of reads”.

8) In section 2.3 Pipeline Steps, number 7 says, “Evaluate the MSA by computing a serie of metrics…”. The word “serie” should be “series”.

9) In section 3.2.2, the sentence “Ths list is avaibale in Table 1” contains three typos. “Ths” should be “The”, “avaible” should be “available”, and there is no punctuation, such as a period.

10) In Table 2, the right-facing triangle to represent the range of values is not canonical. The way to represent a range is to use “en dash”, not a hyphen or “em dash” with no spaces on either side and to use all digits. For example, “2–10” is correct, not “2 - 10”.

11) In the “influence of the sequencing error profile” section, please change “understanding the error rates impact…” to “understanding the error rate impact…”. Also, change “following a ONT error distribution” to “following an ONT error distribution”.

12) For Table 4, why is standard deviation not included in the average difference observed? Frequency should be presented as a percentage (%). Also, why not identify what the best and worst tool are?

Experimental design

1) In the 2.3 Pipeline Steps section, step 2 is “Starting Position Selection”. The default is to select 10 random start positions. How random is the selection of these starting positions? For example, in step 3, genomic regions are constructed with sizes of 100, 200, 500, 1000, 2000, 5000, and 10000. Are the start positions 10,000 bases apart? Will there be overlap between them? What if an end user is only interested in mapping to a gene that is 15,000 bases long? Will the overlap matter? Why even do this step as a randomized process? If it is totally random, can you justify the need for that approach as opposed to, let’s say, a strategy that reflects the length of the reference sequence?

2) In the 2.3 Pipeline Steps section, step 4 is “Read Selection”, where for each region, a set of reads are selected that satisfy the desired depth. If this step wasn’t done, all the smaller reads would theoretically still contribute to the consensus built by the next longest selection of reads. For example, if a consensus is built out of 10 reads that are 10,000 bases long, but the total depth of all reads together is 60x and they all contribute to the full 10,000 base consensus, the quality of the consensus sequence would be much greater in theory. As mentioned in the beginning of the manuscript, “Current computational tools attempt to manage this noise by leveraging redundancy to sift through erroneous bases and accurately represent genomes. One common approach involves multiple sequence alignment…”. This is true. However, by minimizing the redundancy by splitting reads into subsections according to length and depth, the noise management will suffer among real biological reads, in general. Therefore, in this context, the performance of MSA tools can, and should, be gauged on the full dataset.

3) Which ONT error model was used for the synthetic MIX datasets? The manuscript just says, “an ONT error model” without a citation. Also, why only use one ONT error model? The error rate profiles for ONT have been reported to vary widely.

4) In section 3.2.2, it was mentioned that Illumina or HiFi reads would need to be assembled to produce a de facto reference sequence. Also, the manuscript includes three “custom” assembly references and only one standard reference sequence. The paper would improve if the distribution of standard reference consensus at least matched that of the custom assembled “reference” sequences used, especially since the one reference sequence had such a lower error rate compared to the others. Does the error rate reflect the sequencing strategy and/or the assembly job among the “custom” reference sequences?

Validity of the findings

1) In the 2.2 Pipeline Steps section, Step 1 is “Read Alignment”. Only MiniMap2 is used as an alignment tool. There are alternatives. How do those alternatives affect the downstream MSA analysis? Can the end user select which alignment tool they would like to use? Similarly, the MSA evaluation metrics are generated be comparing the consensus sequence to the reference using Exonerate in exact global alignment mode. While MiniMap2 and Exonerate are well known, the results of this paper can only really be interpreted in the context of using those upstream and downstream tools. What about end users that aren’t using those? Will the results still be valid? How will they change and stay the same? This is a concern about generalizability.

2) Why only include datasets of only insertions, only deletions, or only substitutions? That doesn’t reflect a real-world scenario and therefore offers very little in terms of proving the worth of MSA Limit applications in the lab. However, as an exploration of MSA Limit performance, it has some merit. It would just be worth acknowledging the motivation for doing this in the text.

Additional comments

1) Regarding the sentence “Current computational tools attempt to manage this noise by leveraging redundancy to sift through erroneous bases and accurately represent genomes”, please consider citing the following relevant tool as an example: 'Annis, Sofia, et al. "LUCS: a high-resolution nucleic acid sequencing tool for accurate long-read analysis of individual DNA molecules." Aging (Albany NY) 12.8 (2020): 7603.'

2) Good selection of MSA tools to analyze. They are commonly used and offer a breadth of strategies.

3) Good range of error rates tested in the synthetic data.

4) While it isn’t surprising that there is a saturation effect of depth on performance, it is curious why certain tools like POA and MUSCLE drop in performance at deep sequencing depths. Why does this happen? Are errors in alignment being introduced that amplify bias with increasing depth? Another important possibility to consider is the actual genome quality. Are there inversions or other large-scale errors that occur at low frequency? As depth increases, their prevalence could increase, thus adding confusion to the consensus sequence. I suppose the same could be true to small-scale errors too. Please discuss.

5) The synthetic data threshold of 10% error rate for consensus deterioration corresponds to what was seen in the biological data, as the human reference sequence was <10% error. This underscores the importance of including more reference sequences in the paper. If that trend continues in biological data, it will be an interesting finding.

6) Insertions and deletions were mentioned to be more difficult to rectify than substitutions. This is likely due to the problems introduced by frameshift. Please consider discussing this and including the following relevant citation: 'Logan, R., Fleischmann, Z., Annis, S. et al. 3GOLD: optimized Levenshtein distance for clustering third-generation sequencing data. BMC Bioinformatics 23, 95 (2022). https://doi.org/10.1186/s12859-022-04637-7';

7) Good section on diploid genomes and heterozygosity.

8) Also, good discussion section. The meta-MSA results shown in table 4 are interesting.

Reviewer 4 ·

Basic reporting

Overall comments:
This manuscript presents an evaluation of the MSA tools for TGS data. The authors have conducted a comprehensive assessment of different multiple sequence alignment (MSA) methods commonly used in managing sequencing errors in TGS data under different scenarios. The automatic pipeline MSA_Limit is well-wrapped and user-friendly. It offers insights into alignment accuracy, time efficiency, and memory utilization, providing a valuable resource for researchers and developers in selecting appropriate MSA tools for specific experimental settings. Although the results are helpful, I think the experiments can be expanded to cover a few more complicated cases or real scenarios.

Major comments:
1. A major missing part in the experiment is the performance of these tools on handling homopolymer regions. The algorithms and strategies of aligning multiple sequences in those regions can significantly affect the error correcting performance in those regions. Indeed, the error rate in those regions can be much higher than other regions. Knowing the performance of different MSA tools in those regions will be valuable for downstream analysis.

2. As the modern TGS platforms claim higher and higher per-base accuracy, it is better to cite relevant papers about the latest developments in Intro. Some of the used error rates might be too high. Similarly, is 20X counted as low coverage for TGS? In a very important application of Nanopore sequencing, you can use it to survey circulating viral strains in remote areas. The coverage for those data can be quite low. Please cite relevant papers about the “low” coverage.


3. The author defined the step of constructing consensus sequences in 2.4. But it requires the formatted input as row-column multiple sequence alignment. For the tested tools, is there any other step involved to get this format? For example, to extract the row-column multiple sequence alignment from Partial Order Graph.
Tools like abPOA also have their consensus modules and generate a consensus sequence as output. Has the author compared the quality of the tool-generated consensus sequences with the consensus defined in this manuscript?


4. a). In experiment
b). In Figure 6, the actual sequencing depths of two datasets "Human" and "BMB yeast" datasets are less than 120x. It would be helpful to indicate the reason for the absence of 150x and 200x in these two figures.
c). Instead of using error bars, the author may consider using scatter plots or boxplots to avoid exceeding the y-axis limit in Figure 6.

5. It is surprising the see such high error rate for HiFi data.
(Table 1). Can you check this and give some explanations?


Minor-major comments:
1. On Page 2, the author introduced several Partial Order Graph (POG) methods, but the transition from discussing POA and its variations to introducing the evaluation topic appears somewhat abrupt. It would be helpful to clarify what is meant by "alternative MSA methods" in this context.
2. The module and step names depicted in Figure 3 lack clarity and contain redundant information already presented in Figure 2. It is recommended to provide more explicit descriptions of each pipeline step and to relocate Figure 3 to the supplementary file.

3. In 2.5, two metrics “Ambiguous character rate” and “Consensus size” are features of the consensus sequences and are indifferent from the reference sequence.
Thus, the statement in the following paragraph “This set of metrics is computed by comparing the consensus sequence to the reference sequence” is confusing.

4. In Figure 5, there is inconsistency in the region size ranges among the four datasets. Specifically, the dataset "BMB yeast" and "E. coli Illumina" do not include a region length of 10,000 bases.

Minor comments
1. In the abstract, the access to MSA_limit and to the experiments are the same and may only be mentioned once.
2. To enhance readability for readers, it is recommended to use lighter colors instead of dark grey for the modules depicted in Figure 1. In addition, it is the last two rows of the Figure is not immediately
3. In 2.1.1 , “The error profile is determined by dataset selection…” -> sequencing platform selection.
4. The author is advised to use '``' (backtick followed by two single quotes) instead of '”' (straight double quotation marks) in the LaTeX file to ensure that the quotation marks are correctly displayed.
5. The capitalization of the tools in Figure 2 is not consistent with the ones used in the manuscript. It is suggested to ensure consistency between the figure and the manuscript.
6. There is an inconsistency in capitalization within the reference section.
7. Next-generation sequencing (NGS) is used more commonly than SGS.

Experimental design

See box 1.

Validity of the findings

See box 1

Additional comments

None.

---

## Round 0.2 · Minor Revisions

Dear Dr. Rohmer and colleagues:

Thanks for revising your manuscript. The reviewers are mostly satisfied with your revision (as am I). Great! However, there are a few issues still to entertain. Please address these ASAP so we may move towards acceptance of your work.

Best,

-joe

·

Basic reporting

Original Review Comment: "In the third sentence of the introduction, it is written: “third-generation sequencing technologies (TGS)”. The abbreviation should come between “sequencing” and “technologies”. Therefore, all subsequent plural references to TGS should be singular, as it is a
singular class of technology. The same is true for NGS. There are different technologies
within the NGS class of strategies. NGS is a singular entity."

Author's Response: "We now use the singular form instead of the plural form for all mentions of NGS and
TGS in the text."

Follow Up Reviewer Comment: You didn't change the location of the abbreviation. Please do so.

Experimental design

No comment. All previous comments have been addressed.

Validity of the findings

No comment. All previous comments have been addressed.

Additional comments

Original Reviewer Comment: "Regarding the sentence “Current computational tools attempt
to manage this noise by leveraging redundancy to sift through erroneous bases and accurately
represent genomes”, please consider citing the following relevant tool as an example: ’Annis,
Sofia, et al. ”LUCS: a high-resolution nucleic acid sequencing tool for accurate long-read
analysis of individual DNA molecules.” Aging (Albany NY) 12.8 (2020): 7603.’

Author's Response: "Thank you for suggesting the reference to Annis, Sofia, et al.’s work on ’LUCS: a high resolution nucleic acid sequencing tool for accurate long-read analysis of individual DNA
molecules.’ After careful consideration, we have concluded that the specific focus of this
paper diverges from the core methodologies and objectives of our current study.

Follow Up Reviewer Comment: It diverges from your core methodologies, but not the objectives. In fact, LUCS is solely focused on "leveraging redundancy to sift through erroneous bases and accurately represent genomes". Did you really carefully consider it? Why not include it as an example of what you are discussing?

Original Reviewer Comment: "Insertions and deletions were mentioned to be more difficult to rectify than substitutions. This is likely due to the problems introduced by frameshift. Please consider discussing this and including the following relevant citation: Logan R, Fleischmann Z, Annis S, Wehe AW, Tilly JL, Woods DC, Khrapko K. 3GOLD: optimized Levenshtein distance for clustering third-generation sequencing data. BMC Bioinformatics. 2022 Mar 20;23(1):95. doi: 10.1186/s12859-022-04637-7. PMID: 35307007; PMCID: PMC8934446."

Author's Response: "We added a sentence to highlight this potential source of the problem with the appropriate citation."

Follow Up Reviewer Comment: "I cannot find in your revised manuscript where you addressed this or included the citation. The manuscript says 'Substitutions are easier to rectify, but the POA family
struggles with high substitution error rates. Insertion and deletion errors are more challenging,
with deletions being slightly more difficult than insertions.' There is no mention of frameshifts or the inclusion of a relevant citation.

Reviewer 4 ·

Basic reporting

The authors have addressed my comments.

Experimental design

No further comments.

Validity of the findings

No further comments.

---

## Round 0.3 · accepted · Accept

Dear Dr. Rohmer and colleagues:

Thanks for revising your manuscript based on the concerns raised by the reviewers. I now believe that your manuscript is suitable for publication. Congratulations! I look forward to seeing this work in print, and I anticipate it being an important resource for groups studying informatics and genomics. Thanks again for choosing PeerJ to publish such important work.

Best,

-joe